# Pediatric Stroke: A Review of Common Etiologies and Management Strategies

**DOI:** 10.3390/biomedicines11010002

**Published:** 2022-12-20

**Authors:** Cameron A. Rawanduzy, Emma Earl, Greg Mayer, Brandon Lucke-Wold

**Affiliations:** 1Department of Neurosurgery, University of Utah, Salt Lake City, UT 84112, USA; 2School of Medicine, University of Utah, Salt Lake City, UT 84112, USA; 3Department of Neurosurgery, University of Florida, Gainesville, FL 32608, USA

**Keywords:** pediatric stroke, arterial ischemic stroke, hemorrhagic stroke, arterial dissection

## Abstract

Pediatric stroke is an important cause of mortality and morbidity in children. There is a paucity of clinical trials pertaining to pediatric stroke management, and solidified universal guidelines are not established for children the way they are for the adult population. Diagnosis of pediatric stroke can be challenging, and it is often delayed or mischaracterized, which can result in worse outcomes. Understanding risks and appropriate therapy is paramount to improving care.

## 1. Introduction

Stroke is a neurologic injury often associated with significant mortality and morbidity. The underlying mechanism is most frequently due to occluded cerebral blood vessels, i.e., ischemic stroke; however, hemorrhagic stroke is an important etiology [1]. Although the incidence of stroke is more pronounced in the adult population, largely due to the cumulative effects of hypertension, diabetes and atherosclerosis, pediatric stroke can result in death or severe disability, diminishing the quality of life and subsequently imposing serious socioeconomic consequences [1]. The incidence of pediatric stroke is approximately 1.2 to 13 cases per 100,000 children under 18, but due to potential misdiagnoses or a lack of clinical suspicion and workup, these estimates are likely low [1,2,3]. The high-cost burden to the patient and healthcare system warrants greater attention and investigation into the causes and outcomes of pediatric stroke; this review attempts to provide an overview of diagnosis and management strategies for this pathology.

## 2. Etiology and Epidemiology of Ischemic Stroke

Pediatric arterial ischemic stroke (AIS), a significant source of pediatric neurologic morbidity, can lead to numerous debilitating consequences: sensorimotor deficits, behavioral problems, intellectual disability, language impairment, and epilepsy [4]. Pediatric AIS is classified by age, as perinatal AIS broadly encompasses 20 weeks of fetal life through day-of-life 28. Perinatal AIS is further classified by the timing of presentation. Childhood AIS encompasses the first month of life and beyond [4]. Additionally, diagnosing AIS in children is frequently delayed [5,6]. Therefore, understanding and identifying risk factors and at-risk populations is vital in improving the diagnosis and management of AIS.

The risk of pediatric AIS maintains an interesting trend, as children under one year of age maintain the highest risk, with a dramatic decline after one year. This risk remains low through mid-adolescence, at which time risk begins to increase [7]. Additionally, black children have been reported to maintain an increased risk of AIS when compared to white children, and this discrepancy persists after controlling for sickle cell disease [7,8]. Male sex is also reported to be associated with a higher incidence of both childhood and perinatal AIS, and this discrepancy persists after controlling for trauma [7,8]. 

In 2014, Lehman and Rivkin compiled a list of risk factors among perinatal AIS.

Perinatal AIS often manifests due to pregnancy-related risk factors, including hypercoagulability and complex circulation interactions. Table 1 illustrates an adapted version of these risks [9]. Congenital heart disease and arteriopathies present in the infant remain risk factors during both the perinatal stage and beyond the first year of life [4,9]. Thrombophilia in both mother and child remains a risk factor during pregnancy and after pregnancy in childhood AIS [4,9]. These risk factors will be discussed subsequently in the setting of childhood AIS.

There are numerous risk factors for AIS in children, and an overview of these causes is outlined in Table 2 [4]. By definition, pediatric AIS is an acute neurological deficit in a child aged 29 days to 18 years with radiographic evidence of cerebral parenchymal infarction in a known arterial territory that corresponds to their clinical symptoms [10]. The Childhood AIS Standardized Classification and Diagnostic Evaluation (CASCADE) criteria are used to categorize strokes according to the underlying causes [10,11], and arteriopathies are recognized as the most significant risk for childhood AIS [11]. These blood vessel abnormalities can be chronic or acute/transient changes, and they regularly appear on vascular imaging [12,13]. Underlying, inborn or acquired etiologic causes of AIS, such as cardiac disease, metabolic disturbances, thrombophilia and rheumatologic diseases such as systemic lupus erythematosus, may or may not present with vascular imaging changes [12,13]. Focal cerebral arteriopathy (FCA) is a descriptive, catch-all term used to broadly describe transient cerebral arteriopathy (TCA) and other arteriopathies that meet the CASCADE designation of AIS [10]. TCA manifests as a focal stenosis or segmental vessel wall narrowing most often in the distal internal carotid artery, proximal middle cerebral artery and circle of Willis [14]. These angiographic findings are more common in children with abnormal lipid profiles and familial cardiovascular disease [15]. The literature suggests associations between TCA and varicella infections [16], acute herpes virus infections [17] and other upper respiratory infections [18], though the impact of the inflammatory pathways in TCAs remains to be fully understood.

There are other, rare causes of pediatric AIS that warrant discussion, including stroke secondary to genetic disorders such as pseudoxanthoma elasticum [19]. Pseudoxanthoma elasticum is a rare genetic disorder that affects elastic fibers in the skin, eyes, gastrointestinal tract and the walls of blood vessels. The most common symptoms for children with pseudoxanthoma elasticum are retinal and vascular lesions or skin papules; however, though infrequent, cerebrovascular infarcts have been reported and can result in severe neurologic impairment and death [19].

Cardiac disease represents another significant risk factor among pediatric AIS, present in up to 30% of AIS cases [12,20]. Acquired and congenital heart diseases resulted in a 16.1-fold increased risk of AIS when compared to the general pediatric population in a study utilizing the Intermountain Pediatric Stroke Database, with single ventricle pathologies being among the highest risk [21]. Additionally, children with cardiac etiologies of AIS present at an earlier age [22,23]. Recurrent AIS among this patient population is also a concern, as one study reports that 27% of patients with cardiac causes of AIS have another event within 10 years. Recurrent AIS is associated with mechanical valves, thrombophilia, anticoagulation treatment and acute infection [24]. In terms of patent foramen ovale (PFO), pediatric AIS may be a result of embolization due to the right-to-left atrial shunt, though a definitive causation linking PFO with AIS is not established [25]. Claiming PFO as the cause of AIS is based on exclusion after thoroughly ruling out all other etiologies, and PFO closure should be considered on an individual basis [25].

Malignancy, sickle cell disease (SCD) and thrombophilia remain other prominent risk factors among pediatric AIS. The risk of AIS among patients with SCD was dramatically reduced following the 1998 STOP trial, illustrating an effective prevention with chronic transfusion therapy and dramatically reducing AIS in this population where resources are available [26]. Transcranial Doppler, where available, is also an effective screening modality [27]. In some studies, thrombophilia has been observed in up to 50% of pediatric AIS cases [28,29]. An association between thrombophilia and AIS has been established in the following biomarkers: protein C deficiency, factor V G1691A, factor II G20210A, MTHFR C677T, elevated lipoprotein A and antiphospholipid antibodies [30]. 

### Presentation, Symptomatology and Management

Challenges arise when it comes to pediatric stroke diagnosis, in part due to a lack of clinical suspicion because of the diverse and non-pathognomonic presentation. Classically, stroke in adults presents with many universally recognized signs and symptoms. The B.E F.A.S.T. (Balance, Eyes, Face, Arms, Speech, Time) protocol not only delineates common stroke symptoms, such as face droop, unilateral arm weakness and speech difficulties, but also stresses the importance of prompt treatment. Pediatric stroke requires the same urgency but lacks similar protocols for universal recognition. Focal weakness and limb and face weakness can be presentations of pediatric AIS [31]. However, the hallmark sign of stroke, acute hemiparesis, is common only in older pediatric patients [32]. Seizures are the primary presenting symptom for younger children, while nonspecific symptoms such as fever, nausea/vomiting, headache and cardio-pulmonary dysfunction permeate all pediatric patients [32]. The lack of specific symptoms makes recognition more difficult. In addition, a broad differential diagnosis for hemiparesis adds to the complexity of diagnosis in pediatric stroke, as there are many stroke-mimicking disorders. Nearly 21–76% of children with acute-onset focal neurological deficits are found to have migraines, focal seizures, demyelinating diseases, conversion disorders and central nervous system tumors, among other possible diagnoses [33]. The rarity of pediatric AIS and subsequent lack of clinical suspicion can delay diagnosis [34]; however, due to its urgent treatment requirements, AIS should be considered in a differential diagnosis for acute-onset neurologic deficit [34].

In a pediatric patient presenting with stroke-like symptoms, magnetic resonance imaging (MRI) is the preferential imaging modality; however, there are challenges to obtaining them in the pediatric population [31]. High-quality scans can be difficult to achieve in children without sedation because of movement artifacts from restlessness in MRI scanners [35].

Perinatal AIS in the neonatal period is managed with supportive measures, including oxygenation and managing dehydration and anemia [36]. Aspirin and anticoagulation are often not pursued due to the low risk of recurrence of neonatal AIS, though these treatments are considered when AIS is due to cardiac disease [37,38]. Additionally, thrombolytics and mechanical thrombectomy are often not considered in neonates following AIS due to the lack of evidence and small size of neonatal arteries [36,39].

Management of childhood AIS includes strategies targeted towards hypertension, hypotension, hyperglycemia, fever, cerebral swelling and seizures. Acute treatment of childhood AIS focuses on neuroprotective management by maintaining cerebral perfusion and reducing metabolic demand from fevers and seizures [4]. Acute treatment with IV tissue plasminogen activator (tPA) and revascularization are revolutionary in the management of adult AIS, though evidence is lacking in terms of translating this care to children. The Thrombolysis in Pediatric Stroke (TIPS) trial offered potential criteria to safely utilize tPA in children, but these criteria have not been widely accepted [4,40]. The hallmarks of both acute and chronic therapies include anticoagulation and antiplatelet medications. Controversy exists in the literature, however, regarding the optimal treatment. This is in part due to the low risk of bleeding in pediatric AIS, while monitoring the risk of acute hemorrhagic conversion and acknowledging the prominent risk of pediatric AIS recurrence [41,42]. In cases of pediatric AIS with a specific disease etiology, management specific to each disease should be considered, i.e., chronic transfusions in SCD.

## 3. Etiology and Epidemiology of Hemorrhagic Stroke

Hemorrhagic stroke accounts for 35–54% of all childhood stroke [7,43,44,45]. This contrasts with its incidence in adults, where hemorrhagic stroke only accounts for 7.5–19% of all strokes [46,47,48]. As previously described, while there are widespread protocols for adult stroke recognition and management, there is a lack of any similar stroke protocols for the pediatric population. This, among other reasons, may contribute to the increased mortality rate of pediatric hemorrhagic stroke, which can be as high as 54% compared to 18–35% in adults [49,50]. Among the causes of hemorrhagic stroke, one study found that 13% of patients were found to have an underlying cerebral aneurysm, 31% had brain arteriovenous malformations, 2.5% had brain tumors, 25% had an undetermined etiology, and 28.5% had other medial and anatomical etiologies (Figure 1) [51].

## 4. Presentation, Symptomatology and Management

The presentation of hemorrhagic stroke will not differ greatly from AIS, and it requires a similar workup. A computed tomography (CT) head scan may be very useful as a diagnostic tool. In addition to a head CT, MRI, specifically both gradient recall echo (GRE) and susceptibility-weighted imaging (SWI), can be used.

While these methods have not been studied in a pediatric stroke population, these MRI sequences are just as sensitive as CT images for detecting blood in adults with this disorder [52]. Given that almost half of hemorrhagic strokes are due to vascular issues, a complete MRI study, including magnetic resonance angiography and venography (MRA, MRV), can be obtained, when possible, in order to look for additional abnormalities or other causes. Management of hemorrhagic stroke varies and depends on the type and severity of the hemorrhage. In a recent study, 48% of patients were treated with medications, and 45% of patients received surgical intervention, with 32% receiving both medications and surgical intervention and 22% of patients receiving neither medications nor surgical intervention. A total of 19% of the patients in this study were intubated and treated in the intensive care unit (ICU) [31]. Patients with an altered mental status should be monitored in the ICU with an intracranial device, while alert patients can be monitored in a less invasive fashion. Neurological exams should be performed often to evaluate the patient for signs of increased intracranial pressure or herniation of the brain. Vital signs should be kept within appropriate limits for the patient’s age [44]. Surgical intervention may be needed in some patients. Surgical intervention for hemorrhagic stroke may include decompressive craniectomy, resection of an arteriovenous malformation with or without placement of an extracranial drain, and endovascular procedures to treat intracranial aneurysms. Medications used to treat hemorrhagic stroke revolve around symptom management, such as antiepileptics and NSAIDs. Endovascular treatment of intracranial aneurysms has grown in popularity. This is in contrast with surgical treatment, which has been performed at a similar rate to historical trends [53].

## 5. Etiology and Epidemiology of Craniocervical Arterial Dissection

Craniocervical arterial dissection is a known cause of pediatric stroke that has been documented in case reports as early as 1974 [54]. Today, it occurs in 2.5 children per 100,000 annually; however, it accounts for 5–25% of all acute ischemic strokes in the pediatric population [55,56]. There is a significant risk of recurrence if it is left untreated, so if there is a clinical suspicion of AIS due to dissection, proper investigation is required. Children with predisposing factors such as genetic or acquired connective tissue disorders (e.g., copper deficiency) may be more susceptible to dissection; however, the absence of these risks does not rule out the possibility of an ensuing vascular insult [54,57]. The risk of dissection due to connective tissue disorders such as Marfan’s and Ehler–Danlos syndrome is not well elucidated, but it is not negligible and can lead to multiple dissections [58]. The male gender is also a reported risk factor for AIS from craniocervical arterial dissection [57].

Dissections can be traumatic or spontaneous in nature. The pediatric population is at serious risk of traumatic injury, which is the leading cause of mortality and morbidity of children in the United States [59], and dissection is an under-recognized consequence of trauma [1,56]. Many of the head and neck injuries that precede dissections are mechanical or penetrating [56]. However, whiplash injuries are another mechanism by which dissections can occur—there have been some reports of strokes in children after amusement park rides [60]. Additionally, while adult dissections are nearly all traumatic in nature, there is in children a substantial incidence of spontaneous dissections without preceding trauma [58]. Approximately 5–20% of children with a spontaneous dissection have an underlying connective tissue or genetic disorder, or bony or vascular anatomic variations [56]. Most spontaneous dissections occur in the anterior circulation vessels, and cases involving the posterior circulation are more likely to be traumatic [58].

## 6. Presentation, Symptomatology and Management

Dissection in the pediatric population most commonly affects extracranial vessels, specifically the vertebral artery and posterior circulation; this is the reverse of trends in the adult population, where carotid dissections are more prevalent than vertebrobasilar ones [56,61,62]. Posterior circulation dissections present with nonspecific symptoms, including headache, dizziness, vomiting, gait disturbances, altered consciousness and double vision [56]. Anterior circulation dissections tend to present with hemiparesis, facial weakness, speech difficulty and seizures [56,57].

An even rarer subtype of vertebral artery dissection is vertebral artery dissecting aneurysm (VADA). Some congenital cervical anomalies may confer an additional risk of developing VADA, such as in patients with pediatric bowhunter syndrome or rotational vertebral artery compression [63]. VADAs are associated with a worse prognosis [63]. In the pediatric population, the literature on the incidence and cause of these dissecting aneurysms is scarce, but they are linked to prior traumatic injury. The presentation is that of a posterior circulation stroke, and management is challenging.

An early and accurate diagnosis of craniocervical arterial dissection is imperative, because if left undiscovered, there is an appreciable risk of recurrence. A recent study by Uohara and colleagues [61] investigated the difference in the recurrence rate of anterior versus posterior circulation stroke in the pediatric population. The results showed that at 3 years, the recurrence rate was 19% in children with posterior circulation stroke and 4% in anterior circulation stroke [61]. Within their cohort, 20.8% of children with dissections had stroke recurrence, which is a significantly higher figure than the rate seen in the adult population. Additionally, long-term effects of strokes in children are significant, and up to 75% of children affected by stroke can experience lasting neurological deficits [58]. This may be linked to misdiagnosis or an unidentified etiology. An extensive workup, increased monitoring and greater frequency of serial imaging may be warranted, especially in children with posterior circulation strokes [61]. 

A confirmed diagnosis of dissection requires angiography and prompt treatment that may include antithrombotic or antiplatelet therapy, hard collars for neck stabilization, and surgery if the etiology is deemed anatomic. [63]. A non-invasive and non-radiating imaging modality is preferable, given the population [58]. Comprehensive first-line imaging recommendations in the workup of pediatric stroke include MRI and MRA with diffusion weighted imaging (DWI), fluid-attenuated inversion recovery (FLAIR), SWI, and MRA of the head and neck [55]. Vertebral artery compression at the level of C1 and C2 vertebrae during head rotation is a risk factor for dissection, so some studies recommend that dynamic imaging be performed [64]. CASCADE and the Pediatric Stroke Study determined that the presence of one of any three specific findings on angiography could confirm the diagnosis of dissection [56,57], and this is outlined in Table 3, adapted from papers by Nash et al. [56] and Stence et al. [57].

Although neck MRA is part of some stroke protocols, Baltensperger et al. [55] found that its utility may be limited. In a series of 681 patients, all of whom underwent neck MRA in addition to DWI, SWI, and circle of Willis MRA, there was only a single case where a cervical abnormality was reported, with all other findings being normal. Given this low yield, they suggest that neck MRA is not necessary as part of a routine pediatric stroke imaging protocol and may be reserved for patients with suspected vertebral artery dissection and posterior circulation stroke [55].

MRA is highly sensitive, but there are instances where MRA is deficient and cannot identify arterial dissections that were confirmed with digital subtraction angiography [65]. A recent study investigated the utility of arterial wall imaging (AWI) in recognizing and differentiating arteriopathic subtypes in children with arterial ischemic stroke [65]. Their findings suggested that AWI may be useful following abnormal vascular findings identified on standard imaging like MRA. They identified trends towards specific enhancing patterns in different AIS etiologies, but further studies are needed to identify the true utility of this imaging modality [65].

In adult stroke patients, abundant randomized clinical trials have produced robust evidence to support well-established guidelines for management; this is not the case in the pediatric population [66]. Standardized treatment recommendations are limited by a lack of evidence, and the latest guidelines from the 2008 American Heart Association (AHA) and American Stroke association management guidelines for children suggest intravenous anticoagulation for 3 to 6 months bridged to oral anticoagulation in patients with extracranial dissection; furthermore, treatment with anticoagulation is not recommended for intracranial dissections [56]. Further studies are needed. The AHA has also suggested that the decision to use thrombectomy in children can follow adult parameters; however, to date, no thrombectomy trials have included individuals under the age of 18 [66]. Therefore, the role of endovascular therapy and its true benefits are unclear [66].

## 7. Conclusions

Overall, stroke in the pediatric population has a lower incidence than in adults, but it represents an important pathology that is associated with significant mortality and morbidity. Neurologic impairments as a consequence of ischemic and hemorrhagic stroke can be devastating in children due to the effects on the quality of life across the inherently increased lifespan in this population compared to adults. Because of the abundance of stroke-mimicking disorders, the non-pathognomonic presentation and variable symptoms, stroke in pediatrics is frequently misdiagnosed due to a lack of clinical suspicion. The severity of this pathology warrants greater attention and investigation, and standardized protocols, equivalent to those that exist for stroke in adults, should be established.

## Figures and Tables

**Figure 1 biomedicines-11-00002-f001:**
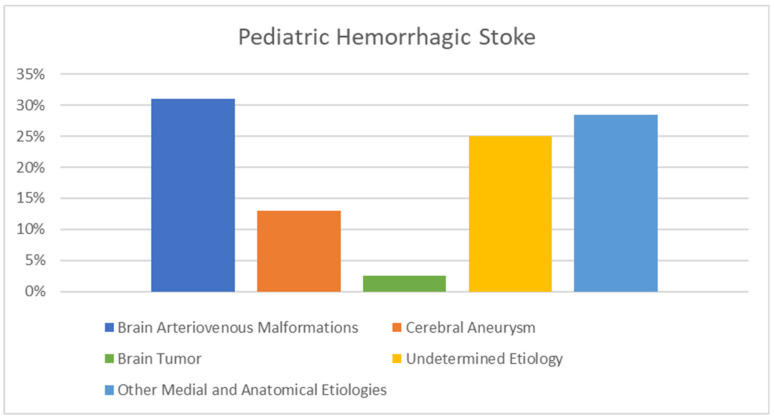
Graph showing the commonality of the different causes of hemorrhagic stroke in pediatric populations. A total of 31% of patients had brain arteriovenous malformations, 13% had a cerebral aneurysm, 2.5% had brain tumors, 25% had an undetermined etiology, and 28.5% had other medial and anatomical etiologies.

**Table 1 biomedicines-11-00002-t001:** Various risk factors for perinatal AIS.

Maternal	Fetal	Placental
Infertility	Hypoglycemia	Chorioamnionitis
PROM	Perinatal asphyxia	Placental infarcts
Pre-eclampsia	Infection	Weight < 10th percentile
Smoking	Resuscitation efforts	
IUGR	5-min APGAR <7	
Infection		
Maternal Fever		

**Table 2 biomedicines-11-00002-t002:** Various risk factors for childhood AIS.

**Arteriopathies**	FCA, TCA	Craniocervical arterial dissection	Fibromuscular dysplasia	Moyamoya disease	Primary CNS angiitis
**Cardiac Disease**	Congenital heart disease	Cardiomyopathy	Arrhythmia	Catheterization, Surgery	ECMO
**Inherited Thrombophilia**	Protein C, S, Antithrombin deficiency	Factor V Leiden	Prothrombin G20210A	MTHFR C677T	Lipoprotein A elevation
**Acquired Thrombophilia**	Antiphospholipid syndrome	Drug-induced			
**Inborn Errors of Metabolism**	Mitochondrial disease	Fabry Disease			
**Rheumatologic Disease**	Systemic lupus erythematosus	Systemic vasculitis			
**Other**	Sickle Cell Disease	Malignancy	Congenital vascular syndromes		

**Table 3 biomedicines-11-00002-t003:** Imaging characteristics of confirmed diagnosis of arterial dissection in pediatrics. Dissection can present in a multitude of ways, and the presence of any one of these findings on angiography represents the occurrence of a dissection.

Diagnostic Confirmation of Pediatric Craniocervical Arterial Dissection		
Double lumen, intimal flap, or pseudoaneurysm identified on angiography, or intramural hematoma (“bright crescent sign”) identified in the arterial wall on axial T1 fat-saturated MRI	Angiographic segmental stenosis or occluded cervical arteries preceded by cervical or cranial trauma, neck pain, or headache <6 weeks prior	Segmental stenosis or occlusion of the vertebral artery occlusion at the level of C2 vertebral body (with or without known history of trauma)

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
