# Peer review of "Pediatric Stroke: A Review of Common Etiologies and Management Strategies"

_biomedicines, 2022, doi:10.3390/biomedicines11010002_

Round 1

Reviewer 1 Report

The authors reviewed pediatric stroke.Article is well written. I have some comments

The title suggests an emphasis on diagnosis and management, but this is not fully reflected in the article

table 2 ; Presentation could be improved

Lines 55-67: found these lines confusing. The authors were following the classification of Felling et al summarized in table 2, which does not correspond to CASCADE classification. It is also important to clarify which categories are descriptive, based on findings on vascular imaging, and which are based on presumed underlying etiology.  The statement "arterial abnormalities detected in up to 80% of children with AIS who underwent vascular imaging" is an over-estimation of arterial pathology as  an etiological factor, even if it is accurate as a radiological finding

Line 90: Presentation and diagnosis were not discussed. in the following paragraphs with no mention of clinical features, suggested imaging or other investigation strategies or expected findings

Lines 116-117@ while delayed diagnosis might contribute to the higher mortality of intracranial hemorrhage in children, this might also reflect difference in underlying etiologies and severity of initial bleeding in this population.  The comment on reasons for delayed diagnosis is mentioned again in lines 131-133, which is a more appropriate location. 

Lines 129-150: these apply to both ischemic stroke and intracranial hemorrhage and could have been described earlier. 

Lines 145-150:  It would be worth mentioning difficulties in obtaining MRI in pediatric population compared to adults and how it impacts diagnostic workout.

Line 168: why was dissection discussed as a separate entity to ischemic stroke and intracranial hemorrhage?

Lines 197-200: the symptoms are not limited to dissection

Table 3: I am not sure why there are three columns. Do they represent different levels of certainty, or different imaging modalities. 

Finally: a conclusion at the end of the article would be helpful. 

Author Response

Reviewer #1:

The authors reviewed pediatric stroke. Article is well written. I have some comments

  • The title suggests an emphasis on diagnosis and management, but this is not fully reflected in the article

RESPONSE: Thank you kindly for your time and consideration while reviewing our manuscript. We agree with the comment that the present title, “Pediatric Stroke: Improving Diagnosis and Management Strategies” may not capture the essence of this review. We have proposed a change to “Pediatric Stroke: A Review of Common Etiologies and Management Strategies” and the manuscript now reflects this change.

  • Table 2 ; Presentation could be improved

RESPONSE: We appreciate you bringing this to attention. We have edited Table 2 within the text to reflect an improved presentation that is simpler to decipher.

  • Lines 55-67: found these lines confusing. The authors were following the classification of Felling et al summarized in table 2, which does not correspond to CASCADE classification. It is also important to clarify which categories are descriptive, based on findings on vascular imaging, and which are based on presumed underlying etiology.  The statement "arterial abnormalities detected in up to 80% of children with AIS who underwent vascular imaging" is an over-estimation of arterial pathology as an etiological factor, even if it is accurate as a radiological finding

RESPONSE: To clarify, the function of table 2 was to serve as a visual summary of the possible risk factors for childhood AIS, it does not represent any classification criteria. We recognize how this paragraph could have led to confusion and have edited the text to more accurately portray our intention. Page 2 Line 50.

Additionally, we agree there is a necessary distincition between descriptive categories and those based on an underlying etiology. To avoid confusion, we have revised the manuscript to define AIS and the CASCADE classification[1] and highlight a distinction between underyling pathology and arteriopathic imaging changes. Page 2 Line 59-90.

  • Line 90: Presentation and diagnosis were not discussed. in the following paragraphs with no mention of clinical features, suggested imaging or other investigation strategies or expected findings

RESPONSE: We have added to the discussion to address the lack of presentation, diagnosis, clinical features, and suggested imaging. Page 3 Line 127-161.

  • Lines 116-117@ while delayed diagnosis might contribute to the higher mortality of intracranial hemorrhage in children, this might also reflect difference in underlying etiologies and severity of initial bleeding in this population.  The comment on reasons for delayed diagnosis is mentioned again in lines 131-133, which is a more appropriate location. 

RESPONSE: We have reformatted the text to introduce this concept earlier. Now; Page 3 Lines 127-133.

  • Lines 129-150: these apply to both ischemic stroke and intracranial hemorrhage and could have been described earlier. 

RESPONSE: We agree with this suggestion. There is redundance and overlap between presentation and management for ischemic and hemorrhagic stroke. We have reformatted the text to describe the symptoms/presentation and suggested imaging earlier under the ischemic stroke subheading. Now, Lines 127-156.

  • Lines 145-150:  It would be worth mentioning difficulties in obtaining MRI in pediatric population compared to adults and how it impacts diagnostic workout.

RESPONSE: Thank you, we have added a discussion of this to the text. Page 4 Lines 152-156.

  • Line 168: why was dissection discussed as a separate entity to ischemic stroke and intracranial hemorrhage?

RESPONSE: With a growing body of evidence regarding dissections and BCVI as a cause of stroke, we included it as its own section. Dissection is an important cause of pediatric stroke that is associated with high recurrence rates. It can be secondary to genetic disorders, anatomic anomalies, or trauma. The pediatric population is very susceptible to mortality and morbidity due to trauma; it is a leading cause of mortality in pediatrics.[2]

  • Lines 197-200: the symptoms are not limited to dissection

RESPONSE: We have reformatted the discussion to introduce this earlier. Now, Page 3 Line 138.

  • Table 3: I am not sure why there are three columns. Do they represent different levels of certainty, or different imaging modalities.

RESPONSE: The columns in Table 3 correspond to the three specific findings on angiography that could confirm the diagnosis of dissection. We have revised the caption of Table 3 to clarify this. Page 7 Line 357-359.

  • Finally: a conclusion at the end of the article would be helpful. 

RESPONSE: We have added a paragraph to conclude the article. Page 7 Lines 380-387.

Reviewer 2 Report

nothing to add

Author Response

NA

Reviewer 3 Report

-The title indicates adequately the study design.  

-The abstract provides an informative summary of the study.

-In the manuscript there is no an adequate and clear presentation of all causes. Rare genetic causes of pediatric stroke, such as PXE, are not reported (Bertamino, Marta, et al. "ABCC6 mutations and early onset stroke: Two cases of a typical Pseudoxanthoma Elasticum." european journal of paediatric neurology 22.4 (2018): 725-728.  DOI: https://doi.org/10.1016/j.ejpn.2018.04.002). Please cite this article.

- Please change the word “pertaining” on the line 27 with a synonimous.

- The manuscript needs an exhaustive conclusion. 

Author Response

Reviewer #3:

The title indicates adequately the study design. The abstract provides an informative summary of the study.

  • In the manuscript there is no an adequate and clear presentation of all causes. Rare genetic causes of pediatric stroke, such as PXE, are not reported (Bertamino, Marta, et al. "ABCC6 mutations and early onset stroke: Two cases of a typical Pseudoxanthoma Elasticum." european journal of paediatric neurology 22.4 (2018): 725-728.  DOI: https://doi.org/10.1016/j.ejpn.2018.04.002). Please cite this article.

RESPONSE: Thank you for your constructive feedback. We have added a discussion of PXE to represent one of the rare, genetic disorders that can lead to stroke in the pediatric population. Page 3 Lines 97-102.

  • Please change the word “pertaining” on the line 27 with a synonimous.

RESPONSE: We have edited out the word “pertaining to” and revised the text. Page 1 Line 27.

  • The manuscript needs an exhaustive conclusion.

RESPONSE: We have added a paragraph to conclude the article. Page 7 Lines 380-387.

References:

  1. Bohmer, M., T. Niederstadt, W. Heindel, M. Wildgruber, R. Strater, U. Hanning, A. Kemmling, and P.B. Sporns, Impact of Childhood Arterial Ischemic Stroke Standardized Classification and Diagnostic Evaluation Classification on Further Course of Arteriopathy and Recurrence of Childhood Stroke. Stroke, 2018: p. STROKEAHA118023060.
  2. Theodorou, C.M., L.A. Galganski, G.J. Jurkovich, D.L. Farmer, S. Hirose, J.T. Stephenson, and A.F. Trappey, Causes of early mortality in pediatric trauma patients. J Trauma Acute Care Surg, 2021. 90(3): p. 574-581.

Round 2

Reviewer 3 Report

Dear authors,

The paper addresses an interesting topic. I think you made good corrections.